# Antibody-Based Therapies for Peripheral T-Cell Lymphoma

**DOI:** 10.3390/cancers16203489

**Published:** 2024-10-15

**Authors:** Nazila Shafagati, Suman Paul, Sima Rozati, Cole H. Sterling

**Affiliations:** The Sidney Kimmel Comprehensive Cancer Center, Johns Hopkins University School of Medicine, Baltimore, MD 21287, USA; spaul19@jhmi.edu (S.P.); srozati1@jhmi.edu (S.R.)

**Keywords:** T-cell lymphoma, immunotherapy, CAR T-cell therapy, bispecific T-cell therapy, CAR-NK cell therapy, bispecific NK cell therapy

## Abstract

**Simple Summary:**

Peripheral T-cell lymphoma (PTCL) is a collection of aggressive malignancies with poor long-term survival and limited effective treatment options. Despite unique challenges associated with identifying and targeting malignant T-cell antigens, the use of antibody-based therapies holds great potential. Here, we review the current and future roles of antibody-based therapies for PTCL.

**Abstract:**

While antibody-based immunotherapeutic strategies have revolutionized the treatment of B-cell lymphomas, progress in T-cell lymphomas has suffered from suboptimal targets, disease heterogeneity, and limited effective treatment options. Nonetheless, recent advances in our understanding of T-cell biology, the identification of novel targets, and the emergence of new therapies provide hope for the future. In this review, we explore four areas of current and evolving antibody-based strategies for the treatment of peripheral T-cell lymphoma (PTCL): monoclonal antibodies (mAbs), bispecific antibodies (BsAs), chimeric antigen receptor T-cell therapy (CAR-T), and antibody–drug conjugates (ADCs). As part of this discussion, we will also include limitations, lessons learned, and potential future directions.

## 1. Introduction

Peripheral T-cell lymphoma (PTCL) represents a rare and heterogeneous form of non-Hodgkin lymphoma with inferior outcomes for most subtypes. According to the 5th edition of the World Health Organization Classification of Lymphoid Neoplasms, there are over 30 subtypes of mature T- and NK-cell lymphoid neoplasms, which can be broadly divided into PTCLs (typically presenting with nodal, extranodal, and/or leukemic involvement) and cutaneous T-cell lymphomas (CTCLs) (typically presenting with cutaneous disease with or without extra-cutaneous involvement) [1]. Subtypes of PTCL and CTCL are outlined in Figure 1. The most common subtype is PTCL, not otherwise specified (PTCL, NOS), followed by nodal T-follicular helper (TFH) cell lymphoma, angioimmunoblastic type (AITL), and anaplastic large cell lymphoma (ALCL). Despite the development of novel therapies and improvements in our understanding of T-cell biology, outcomes remain poor, with 5-year overall survival rates of 30–40% [2].

While antibody-based therapies have revolutionized our management of B-cell lymphomas, their role in PTCL remains to be determined. In addition to disease rarity and heterogeneity, the delayed progress in improving PTCL outcomes is driven in part by fewer targets that can be safely and effectively leveraged in clinical practice. For example, in contrast to the more modest infectious risks seen after B-cell depletive therapy, targeting universal T-cell antigens can result in dramatically immunocompromised states with unacceptable risks of infection and toxicity. Furthermore, many of the antibody-based therapies that are used for the treatment of B-cell lymphomas are unsuitable for T-cell malignancies. This is driven largely by the overlapping targets and functions uniquely shared by both the malignant and effector T-cells implicated in these therapies. For example, antibodies targeting PD-1 and PD-L1 block essential immune inhibitory molecules, leading to augmented T-cell responses. Although increasing T-cell response and proliferation is ideal for effector T-cells involved in neoplastic surveillance and destruction, clonal T-cell expansion and associated hyperprogression of T-cell malignancies is a well-characterized potential complication [3]. More recently, a variety of chimeric antigen receptor T-cell therapy (CAR-T) strategies have been used for the treatment of T-cell lymphomas. While initial responses are often seen, many of these approaches have led to unwanted activation of the malignant T-cells upon engagement with the effector CAR T-cells, resulting in CAR-T destruction and lack of sustained responses.

Despite these limitations, a concerted effort is underway to identify novel targets and treatment options more suitable for T-cell malignancies. Here we review antibody-based strategies in PTCL, including monoclonal antibodies (mAbs), bispecific antibodies (BsAs), CAR-T, and antibody–drug conjugates (ADCs). As part of this discussion, we also include limitations, lessons learned, and potential future directions.

## 2. Monoclonal Antibodies

First developed nearly five decades ago, monoclonal antibodies (mAbs) play an essential role in the treatment of lymphoma. Ranging from well-established B-cell targets such as CD20 and CD19 to targets including CD52 and CCR4 in T-cell malignancies, mAbs continue to expand the treatment landscape in lymphomas [4]. In addition to their role as immune checkpoint inhibitors, monoclonal antibodies exhibit anti-tumoral activity through several known mechanisms of action. One such mechanism is antibody-dependent cellular cytotoxicity (ADCC), in which direct binding of the target cell (via the Fab region of the antibody) with immune cells (namely, natural killer (NK) cells and macrophages) leads to the secretion of cytotoxic granules that induce the RAS system, ultimately leading to cell death. Another such pathway is antibody-dependent cellular phagocytosis (ADCP), where the antibody binding to the target cell induces immune cells to consume the entire cell-antibody complex. Additional mechanisms include complement-dependent cytotoxicity (CDC), wherein antibody binding leads to activation of the complement cascade [4]. More recently, disruption of chemokine-based pathways has demonstrated efficacy, as seen in therapeutic agents such as mogamulizumab. This CCR4-targeting therapy, which will be discussed in depth below, inhibits chemokine-mediated migration and proliferation of T-cells as well as chemokine-driven angiogenesis [5]. Strengths of mAb therapy include targeted action, versatility, and a relatively favorable toxicity profile. Challenges broadly fall into three categories: (1) host-related factors (such as effector cell exhaustion); (2) antibody-related factors (such as antigen affinity); and most importantly, (3) tumor factors (such as antigen expression and tumor microenvironment) [4,6]. The use of mAbs in T-cell lymphomas is further complicated by the inability to safely target pan T-cell antigens and the potential for malignant clonal expansion following immune checkpoint inhibition [4,7]. Nonetheless, mAbs play an important role in the management of PTCL, including rare subsets and settings where there are few remaining viable options.

### 2.1. CD52

T-cell prolymphocytic leukemia (T-PLL) is a rare T-cell malignancy associated with a poor prognosis and limited effective treatment options. Alemtuzumab, a recombinant DNA-derived IgG1 kappa mAb targeting CD52, has demonstrated meaningful efficacy in T-PLL, leading to responses in more than 2/3 of patients [8,9,10]. Although most patients who do not receive consolidative allogeneic stem cell transplantation (alloSCT) eventually relapse, this impressive response rate with otherwise limited effective alternatives has resulted in the NCCN recommendation for alemtuzumab to be the preferred front-line therapy for T-PLL [11], despite lack of FDA approval for this indication [9,10,11]. Alemtuzumab has also been studied in adult T-cell leukemia/lymphoma (ATLL). In a Phase II study including 29 patients with ATLL, 15 responded (51.7%) with a median response duration of 14.5 months. The median progression-free survival (PFS) was 2.0 months, and the median overall survival (OS) was 5.9 months [9]. As CD52 is also expressed on the surface of non-malignant mature lymphocytes, treatment with alemtuzumab and associated lymphodepletion can lead to life-threatening infectious complications, including reactivation of cytomegalovirus (CMV), JC-virus-related progressive multifocal leukoencephalopathy (PML), fungal infections, and non-opportunistic infections [9,10]. Nonetheless, alemtuzumab continues to play a vital role in the treatment of patients with T-PLL.

### 2.2. CCR4

Mogamulizumab is a humanized IgG1 mAb targeting chemokine receptor 4 (CCR4), which is expressed in many T-cell lymphomas as well as most Th2 and regulatory T-cells. While CCR4 expression varies in PTCL, ranging from 30–65% in most subsets [12], it is found in approximately 90% of patients with ATLL. In addition to mycosis fungoides/Sezary syndrome (MF/SS), mogamulizumab has been highly effective in ATLL, resulting in overall response rates (ORRs) of approximately 50% [13]. Compared to the historic median overall survival (OS) of 4 months for patients with relapsed or refractory (R/R) ATLL, treatment with mogamulizumab has resulted in meaningful improvement with a median OS of 13 months. These results led to the approval of mogamulizumab for R/R ATLL in Japan in 2012 [5]. While not FDA-approved for the treatment of ATLL in the United States, mogamulizumab is included as a recommended treatment option for R/R ATLL on the NCCN guidelines [11]. Mogamulizumab has also been studied in other subsets of PTCL. Among non-ATLL forms of R/R CCR4+ PTCL, response rates vary, including an ORR of 11–34% with stable disease (SD) or better in 46–65% [14]. Of note, responses have been seen in PTCL-NOS, AITL, and ALK-ALCL. It remains unclear to what extent response rates correlate with CCR4 expression, and further studies evaluating this relationship may prove beneficial [14,15]. Clinical trials evaluating combination therapies with mogamulizumab are now underway in T-cell lymphomas, including its combination with chemotherapy, brentuximab vedotin (Bv), and allogeneic NK cells.

### 2.3. CD38C

CD38, a well-established target in multiple myeloma therapy, has also been studied in certain subsets of T-cell lymphomas, most notably extranodal NK T-cell lymphoma (ENKTL). In one study involving 94 patients with newly diagnosed ENKTL, CD38 was expressed weakly in 47 patients and strongly in 47 patients, with higher CD38 expression associated with a worse prognosis [16]. This finding has led to studies investigating the role of daratumumab, an IgG1 kappa CD38-directed mAb often used for the treatment of multiple myeloma, in the treatment of patients with PTCL. Unfortunately, daratumumab monotherapy for ENKTL has led to only modest response rates. In a Phase II study including 32 patients with R/R ENKTL, treatment with daratumumab led to an ORR of 25% with no complete responses (CRs) [12,17]. One possible explanation for the reduced response is the upregulation of the complement inhibitory proteins (CIPs) CD55 and CD59, which have been associated with later-stage ENKTL and can suppress the potency of daratumumab-induced CDC [18].

### 2.4. CD30

CD30 is a transmembrane receptor in the tumor necrosis factor (TNF) receptor family, which is universally expressed in classic Hodgkin lymphoma and ALCL, as well as most other PTCL subtypes, with variable expression in approximately 55–60% of cases [19,20]. As will be discussed later in this review, CD30-targeted therapy has dramatically impacted the treatment of PTCL through the CD30-directed ADC brentuximab vedotin. However, prior to the development of brentuximab, the CD30-directed mAb SGN-30 was evaluated in patients with CD30+ lymphomas. Despite demonstrating safety, monotherapy response rates were low, and efforts were ultimately abandoned to prioritize the ADC form [21,22].

### 2.5. PD-1/PD-L1

Monoclonal antibodies targeting programmed cell death protein 1 (PD-1) and programmed death-ligand 1 (PD-L1), collectively referred to as immune checkpoint inhibitors (ICIs), have revolutionized the treatment of solid tumors but have also demonstrated efficacy in hematologic malignancies, including T-cell lymphomas. PD-L1, expressed in varying degrees on the surface of neoplastic cells, binds PD-1 on T-cells, inhibiting cytotoxic T-cell mediated cellular killing and leading to immune evasion. Targeting PD-1 or PD-L1 disrupts this interaction and augments T-cell-mediated destruction of malignant cells. In the largest prospective study to date of PD-1/PD-L1 targeted therapies in PTCL, 102 patients from 41 centers in China were enrolled to receive the anti-PD-1 antibody geptanolimab for R/R PTCL. Among 89 patients included in the analysis, the ORR was 40.4%, with a complete response rate (CRR) of 14.6%. Compared to those with PD-L1 expression <50%, those with PD-L1 expression ≥50% had a better ORR of 53.3% vs. 25.0% and improved median progression-free survival (PFS) of 6.2 months vs. 1.5 months. On subgroup analysis, the ORR was best in ENKTL (63.2%, 12/19), followed by ALK+ALCL (53.8%, 7/13), ALK-ALCL (42.9%, 3/7), and PTCL-NOS (17.9%, 5/28) [23]. Similar results have been seen with pembrolizumab [24] and nivolumab [25].

One option for improving response is to combine ICIs with agents that increase antigen expression for immune priming. In a Phase II study among 28 patients with R/R T-cell lymphomas, the combination of romidepsin, a histone deacetylase inhibitor (HDACi), and pembrolizumab led to an ORR of 39.5% with a CRR of 34.2%. The combination was generally well-tolerated despite three patients discontinuing therapy early due to immune-related adverse events (iRAEs) [26]. Another approach is to incorporate ICIs into consolidation; in a Phase II study, among 21 patients with PTCL in first remission undergoing autologous stem cell transplantation (ASCT), the estimated 18-month PFS was 83.6% and OS 94.4%, with 14/21 patients completing the planned eight cycles of pembrolizumab post-ASCT. The safety profile was as expected with pembrolizumab, and there were no grade 5 toxicities [27].

Among T-cell lymphomas, PD-1/PD-L1 blockade appears to be most promising in patients with ENKTL. When Jo et al. performed immunostaining and retrospectively analyzed medical records in 79 patients with ENKTL, PD-L1 was expressed in 79.7% of tumor cells, and PD-1 was expressed in only 1.3%. In this study, comparing individual biopsy results with clinical outcomes demonstrated a trend toward better OS in patients with PD-L1 positivity on tumor cells [28]. In a Phase II trial, the anti-PD-L1 antibody avelumab was given to 21 patients with R/R ENKTL, resulting in an ORR of 38% and CRR of 24%. Responses correlated with tumor PD-L1 expression [29]. Treatment with the anti-PD-1 monoclonal antibody pembrolizumab resulted in similar responses in patients with relapsed or refractory disease [30]. In a single-arm, multicenter Phase II study involving 80 patients with relapsed or refractory ENKTL, the anti-PD-L1 monoclonal antibody sugemalimab resulted in an ORR of 44.9% and CRR 35.9% with a 12-month duration of response rate of 82.5% among responders [31]. Efforts are now underway to incorporate anti-PD-1/PD-L1-based therapy into earlier lines of therapy for ENKTL.

Aside from its role in ENKTL, therapeutic immune checkpoint inhibition in PTCL remains limited due to underwhelming efficacy and concerns about inducing hyperprogression. As malignant T-cells may express either PDL-1 or PD-1, treating with immune checkpoint inhibitors can also lead to the expansion of clonal T-cells and associated disease progression. In a 2018 study of ATLL patients with increased mutational load and overexpression of PD-L1 treated with nivolumab, the first 3/3 patients developed hyperprogression after a single dose of nivolumab, resulting in early study discontinuation [7]. Similarly, in a 2022 Phase II prospective study of single-agent nivolumab for R/R PTCL, 4/12 patients demonstrated rapid progression of their disease, 3 (75%) of which had angioimmunoblastic T-cell lymphoma (AITL) [25]. Interestingly, evidence from mouse models of T-cell lymphoma suggests PD-1 may act as a tumor suppressor, providing a possible explanation for the hyperprogression seen in these trials [32]. This theory is further supported by real-world data in humans [33].

### 2.6. Future Directions

Future directions for the use of mAbs in PTCL include novel targets and combination therapies. One of these novel targets is CD94, a protein expressed on NK cells and CD8+ T-cells [34], which is now being targeted in a Phase I trial treating large granular lymphocytic (LGL) leukemia and cytotoxic T-cell lymphomas. Other targets being treated with investigational mAbs in PTCL include TNF receptor 2 (TNFR2), a signaling molecule found on the surface of regulatory T-cells that mediates proinflammatory responses [35]; the inducible T-cell costimulator (ICOS), which is highly expressed by T-follicular helper cells (TFHs) [36]; and the killer cell immunoglobulin-like receptor (three domains) long cytoplasmic tail 2 (KIR3DL2) to improve NK cell recruitment and ADCC [37].

In addition to novel targets, new combinations incorporating monoclonal antibodies are an appealing option. For example, combining immunomodulatory therapies such as lenalidomide with mAbs can augment ADCC. This approach has improved outcomes in B-cell malignancies when combined with rituximab [38], obinutuzumab [39], and daratumumab [40] and is now being evaluated with the anti-PD-L1 monoclonal antibody durvalumab in T-cell lymphomas [41]. Additional combinations involving mAbs for PTCL under investigation are described in Table 1. Mechanisms of antibody-based treatment modalities with selected examples for PTCL are shown in Figure 2.

## 3. Bispecific Antibodies

Bispecific antibodies (BsAbs) have emerged as an effective and well-tolerated, “off-the-shelf” alternative to chimeric antigen receptor T-cell therapy (CAR-T). Through simultaneous binding of CD3 on T-cells and a target antigen on malignant cells, bispecific T-cell-engaging therapies (TCEs) engage the patient’s T-cells for targeted anti-neoplastic activity. In addition to seven FDA-approved BsAbs for the treatment of B-cell malignancies [42,43,44,45,46,47,48], these therapies are now being used in solid malignancies as well, including uveal melanoma [49], small cell lung cancer [50], and EGFR exon 20 insertion-mutated non-small cell lung cancer [51]. The use of TCEs in T-cell malignancies has been more limited for several reasons. First, targeting pan T-cell antigens or those frequently expressed in all healthy T-cells—such as CD3, CD5, and CD7—can lead to T-cell aplasia, rendering patients profoundly immunocompromised and at risk for life-threatening infections [52]. In addition, the engagement of an effector T-cell with a target (malignant) T-cell often results in bidirectional T-cell destruction. This phenomenon has been well described in CAR-T, but is also seen in bispecific antibodies targeting T-cell malignancies.

The ideal BsAb target would consist of an antigen expressed on all malignant T-cells but not all healthy T-cells. One option is to target one of two T-cell receptor (TCR) β chain constant regions (TRBC1 or TRBC2), which should allow for the depletion of all malignant T-cells while sparing ~50% of healthy T-cells. However, when Paul et al. used a TCE targeting TRBC1-expressing T-cell cancers in vitro, both the malignant T-cells and most healthy human T-cells (including TRBC1+ and TRBC2+ T-cells) were depleted, consistent with unwanted bidirectional killing. A more specific approach is to target one of thirty TCR β chain variable gene families (TRBV1 to TRBV30). In the same study, using TRBV-targeted TCEs depleted cancerous T-cells in vitro and in vivo while preserving most normal T-cells. However, this approach is limited by practical considerations, as it would require the development of 30 distinct therapies—each targeting a different TRBV gene family—to treat all mature T-cell lymphomas [53]. As small malignant subclones have been identified in PTCL [54], there is also the question as to whether small malignant subclones may not share the same variable gene family and could thus escape untargeted. As CD30 is expressed in most T-cell lymphomas and has been safely and effectively targeted with the ADC brentuximab vedotin [55], there are efforts being made to develop a CD30-directed TCE. A first-in-human study is now underway evaluating anti-CD30/CD3 BsAb-coated activated patient T-cells (ATCs) ex vivo before autologous re-infusions. This approach aims to harness the specificity of the BsAb with the power of cellular therapy [56]. Another protein that has been targeted with a TCE in animal studies is CD70, a type 2 transmembrane protein of the TNF family, which is expressed by many malignancies including many T-cell cancers. Despite the potent killing of CD70+ cancer cells in vitro and in vivo, toxicology studies in cynomolgus monkeys suggest a potential for unanticipated therapy-related injury in mesothelial and epithelial cells despite very low target antigen expression [57].

While the role of TCEs in T-cell malignancies remains to be determined, other bispecific antibodies have shown promise, particularly the CD30/CD16a bispecific antibody, AFM13. As CD16a is expressed on NK cells and macrophages, AFM13 is considered a tetravalent, bispecific Innate Cell Engager (ICE), allowing for the destruction of CD30-expressing malignant cells through targeted ADCC. In a Phase II study of AFM13 monotherapy in patients with CD30+ R/R PTCL, among 108 patients, the ORR was 32.4% with a CRR of 10.2%. Responses were seen in a variety of subtypes and were highest in AITL (n = 30), where the ORR was 53.3% [58]. In a follow-up Phase I/II study of R/R CD30+ lymphomas, treatment with AFM13 in combination with pre-complexed cord blood (cb)-derived allogeneic NK cells that were IL-12/IL-15/IL-18-preactivated and subsequently ex-vivo expanded resulted in an ORR of 94% and CRR of 71% for those treated at the recommended Phase II dose (R2PD) (n = 36); however, the vast majority of patients had classical Hodgkin lymphoma (cHL) rather than PTCL [59]. A Phase II study (LuminICE) is currently underway assessing AFM13 in combination with the allogeneic cord blood (cb)-derived NK cell product optimized for enhanced ADCC through selection for the KIR-B haplotype and CD16 F158V polymorphism (AB-101) for the treatment of R/R CD30+ cHL and PTCL [60]. Another BsAb being investigated in T-cell lymphomas is the anti-PD-1/CD3 BsAb ONO-4685, which aims to augment T-cell-mediated destruction of malignant cells. Current clinical trials evaluating BsAbs in PTCL are shown in Table 2.

## 4. CAR T-Cell Therapies

CAR T has dramatically changed the management of B-cell malignancies by targeting antigens, such as CD19 for B-cell lymphomas [61,62,63] and B-cell maturation antigen (BCMA) for multiple myeloma [64]. This approach imparts the ability for healthy T-cells to recognize malignant cells through an engineered chimeric antigen receptor (CAR), which recognizes antigens expressed on malignant cells with the same specificity as an antibody but with the killing power of T-cells. Utilizing this foundational technique, CAR-T trials targeting CD3, CD5, CD7, TRBC1, and CCR4 are ongoing and possess immense potential in the treatment of T-cell lymphomas. As described in the bispecific antibody section of this review, however, the barriers to relying on effector T-cells for the killing of malignant T-cells through targeted approaches have been well-characterized [52], including healthy T-cell aplasia, fratricide, or unwanted killing of CAR T-cells, and specifically for CAR-T, contamination of pheresis product with malignant T-cells. In this section we discuss ongoing trials and future directions of CAR-based therapies for the treatment of PTCL.

### 4.1. CD7

Perhaps more than any other antigen, CD7 has been targeted in numerous CAR-T studies for T-cell malignancies. While CD7 is expressed on T-cell acute lymphoblastic leukemia (T-ALL) blasts and most mature T-cell lymphomas, it is also expressed on most healthy T-cells, posing therapeutic challenges due to fratricide and T-cell aplasia. Various approaches have been successfully used to mitigate fratricide in this setting. For example, targeted disruption of the CD7 gene using cluster regularly interspaced short palindromic repeats (CRISPR)/CRISPR-associated protein 9 (Cas9) prior to CAR expression has been found to reduce fratricide and improve expansion of CD7-knock out (KO) CD7-CAR T-cells in preclinical models [65]. The use of CRISPR-induced base editing of CAR T-cells has since been used in clinical trials, including a Phase I study of pediatric T-ALL patients treated with off-the-shelf anti-CD7 CAR-T with inactivated CD52, CD7, and β chain of the αβ T-cell receptor to evade lymphodepleting serotherapy, fratricide, and graft-versus-host disease, respectively [66].

Another option is selecting naturally occurring CD7-negative T-cells during apheresis, creating a population of anti-CD7 CAR^CD7−^ T-cells [67]. Although this obviates the need for resource-intensive genetic editing, given the relative rarity of CD7-negative T-cells, collecting sufficient functional T-cells needed to manufacture CD7-directed CAR-T remains challenging. The use of “naturally selected” CD7-CAR T-cells (NS7CAR) without genetic manipulations was studied in a Phase I clinical trial for T-ALL/lymphoblastic lymphoma (LBL). Among 20 patients treated with NS7CAR-T, 19 achieved minimal residual disease-negative complete response (CR) in the bone marrow by day 28, and 5 of 9 achieved extramedullary CR [68]. Another CD7-CAR-T evaluated in clinical trials includes the off-the-shelf, allogeneic, genetically modified anti-CD7 CAR-T product, WU-CART-007. In a Phase I/II study presented at the 2024 European Hematology Association (EHA) Congress, among 11 evaluable, heavily pretreated patients who received WU-CART-007 at the recommended Phase II dose (R2PD), the ORR was 91% and CRR 82%. With a median follow-up of 2.7 months, 6/11 patients remained in continuous CR at 1.3 to 4.3 months [69].

Alternative strategies for successful CD7-CAR-T deployment with fratricide mitigation include the integration of inhibitors of key CAR/CD3ζ signaling kinases. In a T-ALL mouse xenograft study, unedited CD7 CAR T-cells supplemented with the tyrosine kinase inhibitors ibrutinib and dasatinib led to robust ex vivo expansion with minimal fratricide, as well as reduced terminal differentiation [70]. More recently, an ”all in one” approach of sequential CD7-CAR-T followed by haploidentical alloSCT was evaluated in 10 patients with R/R CD7-positive hematologic malignancies. Following CAR-T, all 10 patients achieved CR with grade 4 pancytopenia. After alloSCT, one patient died due to septic shock, eight achieved full donor chimerism, and one had autologous hematopoiesis. With a median follow-up of 15.1 months after CAR-T, six patients remained in minimal residual disease (MRD)-negative CR, with an estimated 1-year OS of 68% and 1-year disease-free survival of 54% [71]. While CD7-CAR-T has primarily been studied in T-ALL, loss of CD7 is relatively common in mature T-cell lymphomas [72,73], which may limit this approach in PTCL. Nonetheless, the use of CD7-CAR-T is now being studied in clinical trials for PTCL as well.

### 4.2. CD5

CD5 is a pan-T-cell marker that is also being targeted with cellular therapies. As with CD7, anti-CD5 CAR-T can lead to the unwanted killing of CD5+ CAR T-cells. In preclinical studies of anti-CD5 CAR T-cells, substituting the CD28 costimulatory domain with 4-1BB led to increased fratricide. These effects were attributed to Tumor necrosis factor receptor-associated factor (TRAF) signaling from the 4-1BB domain and associated upregulation of the intercellular adhesion molecule 1, leading to stabilized fratricidal immunologic synapses between CD5 CAR T-cells [74]. When compared to wild-type CD5-CAR-T, CRISPR-Cas9-based KO CD5-CAR T enhances anti-tumor efficacy with improved CAR-T activation and proliferation, which may be driven in part by reduced fratricide [75]. Dai et al. developed CRISPR-Cas9-based CD5 KO T-cells targeting CD5 with bi-epitopic CARs using heavy-chain-only antigen recognition domains. When tested in vitro and in vivo with mouse models, this approach led to potent anti-tumor effects with limited fratricide [76]. Hill et al. conducted a Phase I dose-escalation study of an autologous CD5-CAR-T product engineered to produce minimal and transient fratricide [77]. Among nine patients with mature T-cell lymphomas who received treatment, the ORR was 44%, with a CRR of 22%. There were no instances of grade 3 or higher cytokine release syndrome (CRS), neurologic events, or major infectious concerns [78].

Concerns for tumor antigen escape have led to the interest in dual-targeted CARs. For example, Dai et al. developed CD5/CD7-CAR T-cells with CD5 and CD7 KO to mitigate fratricide with promising results [79]. Another option is to use sequential CAR-T to address antigen escape. In a Phase I study of donor-derived CD5-CAR-T in patients with R/R T-ALL, 10 of 14 patients enrolled had CD7 negativity or low expression (including 7 post-CD7-CAR-T and 3 CAR-T-naive). Among 12 patients who received treatment, the day 30 CRR was 100%. With a median follow-up of 10.9 months, two of three patients who underwent consolidative alloSCT maintained remission, with one who died of complications related to the transplant. Of the nine patients who did not pursue consolidative alloSCT, two maintained remission, three relapsed with CD5 loss, and three died from late-onset grade 5 infections (two with EBV and one with Staphylococcus haemolyticus). Of note, all three patients who died of infectious complications were found to have predominantly CD5^-^CD7^-^ T-cells, suggesting the potential for increased risk of severe infectious complications in patients treated with both CD5- and CD7-targeted therapy, whether administered simultaneously or in sequence [80].

In addition to methods for minimizing fratricide in CAR-T, novel options for reducing T-cell aplasia have also been studied. Patel et al. used CRISPR-Cas9 to develop a unique autologous CD5 KO anti-CD5 CAR T product consisting of >90% CD5 KO and >30% CAR transduction to generate two populations of cells: one with CD5 KO CAR-T (to target malignant T-cells) and one with CD5 KO normal untransduced cells (to reduce toxicity from healthy T-cell aplasia) [81]. This approach has been studied in mice and is now being evaluated in a Phase I clinical trial.

### 4.3. CD3

CD3 is a T-cell antigen that is universally expressed in both healthy and malignant mature T-cells. As with CD7-CAR-T, unmanipulated anti-CD3 CAR-T leads to widespread fratricide, limiting the use and efficacy of CAR T. The use of a CD3 protein expression blocker (PEB) composed of an anti-CD3 single-chain variable fragment and an intracellular retention domain to downregulate CD3 has been used to generate a CD3 protein expression blocker (PEBL-CAR T-cells). In preclinical studies involving immunodeficient mice engrafted with CD3+ Jurkat cells, CD3 PEBL-CAR-T has successfully limited fratricide and demonstrated anti-tumor effects with a marked reduction in leukemic cell burden [82]. Another option for targeting CD3 while avoiding the destruction of CD3-expressing CAR T-cells is the use of anti-CD3 CAR NK cells. In a 2016 study by Chen et al., anti-CD3 CAR NK cells were engineered using an NK cell line (N92). CD3CAR N92 cells were found to effectively destroy CD3+ human PTCL samples and T-cell leukemia cell lines ex vivo and controlled Jukrat tumor cell growth in vivo, leading to prolonged survival in mice [83]. These approaches have not yet been evaluated in clinical trials.

### 4.4. CD4

As most PTCLs express CD4, this represents another target of interest. Attempts to treat T-cell lymphomas with anti-CD4 directed therapy date back to at least 1996, starting with chimeric mAb therapy [84]. After the humanized anti-CD4 mAb zanolimumab was studied in Phase II trials in both CTCL [85] and PTCL [86] with inadequate response rates, efforts have since turned to anti-CD4 CAR therapies. Pinz et al. developed a CD4-directed CAR T-cell product in which CD4-CAR T-cells maintained a central memory stem cell-like phenotype (CD8+CD45RO+CD62L), demonstrated anti-tumor responses against CD4+ lymphoma cells, and led to prolonged mouse survival [87]. As with CD3, anti-CD4 CAR NK cells using N92 have also been engineered and demonstrated to lead to responses ex vivo and in vivo with mouse studies [88]. Clinical trials using CD4-CAR-T for PTCL are now underway.

### 4.5. CD30

CD30-directed CAR-based therapies are an enticing option due to high malignant cell expression with more limited healthy cell expression [89]. In vitro efforts to target CD30 with CAR-T date back to the 1990s [90]. Since then, numerous clinical trials have evaluated the role of CD30-directed CAR-T, primarily for the treatment of classical Hodgkin lymphoma but also ALCL and other T-cell lymphomas in ongoing clinical trials. Despite ORRs of 33–39% in initial studies for R/R cHL [91,92] and as high as 72% in patients receiving fludarabine-based lymphodepletion prior to CAR-T infusion [93], the majority of patients have relapsed within 1 year of treatment with 1-year PFS of 36% or lower. Of note, in studies evaluating anti-CD30 CAR-T from in vitro cells to in vivo mouse models, third-generation CAR-T containing two costimulatory domains (CD28 and 4-1BB) demonstrated superior anti-tumor activity and tumor honing ability compared with second-generation CAR-T containing only one costimulatory domain (CD28). Benefits were attributed primarily to increased cytokine secretion with IFNγ and associated low-exhausted phenotype, with the authors emphasizing the importance of 4-1BB signaling for CAR-T persistence [94]. In addition to its use in isolation, anti-CD30 CAR-T has also been investigated as consolidation following autologous stem cell transplantation using carmustine, etoposide, cytarabine, and melphalan (BEAM) conditioning in patients with high-risk CD30+ lymphomas. Among 18 patients who received treatment (11 with cHL, 6 with T-cell lymphoma, and 1 with grey zone lymphoma), at a median follow-up of 48.2 months, the median PFS was 32.3 months, and the median OS was not reached [95]. CD30 CARs have also been added to EBV-virus-specific T-cells, generating off-the-shelf allogeneic CD30-CAR EBVSTs capable of killing both CD30+ lymphoma cells and alloreactive T-cells [96]. Previously studied in cHL, this approach is now being investigated in other CD30+ lymphomas, including PTCL.

### 4.6. CD70

CD70, as discussed above, is a type 2 transmembrane protein of the TNF family, which engages with its ligand CD27 to serve as a costimulatory signal leading to lymphocyte activation. CD70 was previously targeted as a monoclonal antibody with good safety and tolerability [97]. In the Phase I COBAL-LYM dose escalation study of allogeneic CD70-CAR T-cells (CTX 130) for R/R mature T-cell lymphomas, among four patients with PTCL treated at the dose level of 3–4 the ORR was 75% [98]. CD70 has also been targeted with CAR NK cells. In a 2023 study by Rafei et al., a novel retroviral vector targeting CD70 was designed that also ectopically produced interleukin-15 to support NK cell proliferation and expressed the suicide gene inducible caspase 9 (iC9), which could be pharmacologically activated to eliminate transfused cells in the event of toxicity. This CAR70/IL-15 construct demonstrated targeted killing of CD70+ cell lines with improved cytokine production and increased expression of markers of cytotoxicity, including granzyme b, perforin, and TRAIL [99].

### 4.7. CCR4

Similar to the mAb mogamulizumab, CCR4-targeted CAR-T is also under investigation. As CCR4 is expressed on limited subsets of healthy T-cells—such as type-2, type-17 helper T-cells, and regulatory T-cells—targeting CCR4 leads to less immunosuppression. In a study by Watanabe et al., fratricidal depletion of CCR4+ T-cells led to a relative increase of CAR T-cells in the final product. These CCR4-depleted anti-CCR4 CAR T-cells enriched in Th1 and CD8+ T-cells led to potent anti-tumor activity against CCR4-expressing T-cell malignancies in mice [100]. As anti-CD30 CAR T-cells expressing CCR4 lead to improved tumor honing and anti-lymphoma activity when compared with anti-CD30 CAR T-cells lacking CCR4 [101], this approach is now being studied in a clinical trial of patients with R/R CD30+ cHL and CTCL.

### 4.8. TRBC1 and TRBC2

As described previously in this review, components of the clonal TCR represent a promising target for cellular therapies. In the Phase I/II AUTO-4 study, Cwynarski et al. studied tumor biopsies from 73 patients with R/R PTCL and identified 26 with TRBC1+ disease. Among nine evaluable patients who underwent pheresis, lymphodepletion, and TRBC1-directed CAR T-cell (AUTO4) infusion, five achieved a CR, one achieved a PR, and three did not respond. Of note, CAR-T expansion was not seen in the peripheral blood by PCR, raising the concern for a limited duration of response [102]. This phenomenon was further supported by studies from Nichakawade et al., in which incubation of anti-TRBC1+ CAR T-cells with normal T-cells and TRBC1+ T-cell cancers led to bidirectional killing, resulting in the depletion of both anti-TRBC1 CAR T-cells and TRBC1+ normal T-cells [103]. TRBC2-directed CAR T-cells have been developed but have not yet been evaluated clinically [104].

### 4.9. γδ TCR

As with components of the αβ TCR, the γδ TCR can also be targeted. Mature T-cell lymphomas arising from γδ T-cells include the vast majority of hepatosplenic T-cell lymphomas (HSTCLs) and primary cutaneous gamma delta T-cell lymphomas (PCGDTCLs), both of which are rare, challenging to treat, and associated with poor prognosis [105,106]. The γδ TCR is also expressed in a substantial proportion of patients with monomorphic epitheliotropic intestinal T-cell lymphoma (MEITL), another rare subset of PTCL with limited effective treatment options, with reports of γδ TCR expression in up to 78% of cases [107]. As γδ T-cells make up less than 5% of healthy peripheral T-cells, targeting the γδ TCR is an attractive treatment option for these patients. Wawrzyniecka developed a CAR-T product in which normal αβ T-cells were engineered to express anti-γδ TCR CAR and demonstrated the killing of the malignant γδ T-cells in vitro and in vivo. As has been seen with other CAR-T efforts in T-cell malignancies, however, CAR T-cell persistence was limited, raising concerns about the bidirectional killing of the effector and target cells [108].

### 4.10. Future Directions

In addition to incorporating novel targets, future investigation into CAR-T for PTCL and other T-cell malignancies will aim to reduce fratricide, improve CAR-T persistence, and address healthy T-cell aplasia. While CAR therapy has traditionally focused on a T-cell effector cell, CAR NK cells have also emerged as an area of interest in a wide variety of malignancies. Unlike CAR-T, CAR NK cell therapy is typically off-the-shelf, associated with lower toxicity (particularly CRS and neurologic toxicity), and does not cause graft-versus-host disease (GVHD) [109]. In the case of T-cell malignancies, utilization of NK cell-based therapies is especially appealing given the reduced risk of fratricide and malignant T-cell contamination of the CAR product. Potential disadvantages of CAR NK-cell therapy include a shorter half-life, questionable persistence and efficacy, and challenges associated with NK cell isolation and efficient transduction [110]. There are currently 15 trials listed on clinicaltrials.gov using CAR NK cells for T-cell malignancies, as shown in Table 3.

## 5. Antibody–Drug Conjugates

Antibody–drug conjugates (ADCs) represent a burgeoning approach to cancer therapy that offers the cytotoxicity of conventional chemotherapeutic agents with the targeted specificity of mAbs. There are three critical components to ADCs: the mAb providing targeted specificity, the cytotoxic payload delivering anti-neoplastic activity, and a linker connecting the mAb and payload. ADCs provide several mechanistic benefits over CAR-T and TCEs, including controlled drug release, fixed pharmacodynamics, reduced risk of immunologic reactions, and ease of manufacturing. For example, CAR T-cell persistence may be considered a double-edged sword; while a single infusion can lead to prolonged disease control, it can also cause persistent or recurrent immunologic toxicity. In contrast, the pharmacokinetics and pharmacodynamics of ADCs are more controlled and predictable, with dosing schedules approximating conventional chemotherapy. ADC manufacturing and administration is also more scalable with off-the-shelf availability and the ability to safely administer in the community without the intensive monitoring for immunologic toxicity required for CAR-T or TCEs [99]. Closely related to ADCs are immunotoxins and radioactive isotope conjugates, which deliver targeted toxins and radioactive particles, respectively, to kill malignant cells. ADCs currently approved for use in lymphomas include brentuximab vedotin (Bv, targeting CD30), polatuzumab vedotin (targeting CD79B), and loncastuximab tesirine (targeting CD19). Of these agents, only Bv is approved in PTCL, but it is arguably the single most important advancement for the treatment of PTCL in decades.

Bv is a CD30-targeted ADC with a monomethyl auristatin E (MMAE) payload, which leads to anti-neoplastic activity through microtubule disruption. In the pivotal Phase III ECHELON-2 study, Bv plus cyclophosphamide, doxorubicin, and prednisone (Bv-CHP) led to improved PFS and OS when compared with cyclophosphamide, doxorubicin hydrochloride, vincristine sulfate (Oncovin), and prednisone (CHOP) for patients with untreated PTCL. Among 452 patients with untreated CD30+ PTCL (≥10% expression by immunohistochemistry [IHC]), with a median follow-up of 47.6 months, the 5-year PFS was 51.4% vs. 43.0% (hazard ratio = 0.70; 95% confidence interval [CI] 0.53–0.91) and the 5-year OS was 70.1% vs. 61.0% (hazard ratio = 0.72; 95% CI 0.53–0.99) for patients receiving Bv-CHP vs. CHOP, respectively. Of note, 70% of patients receiving treatment had ALCL as prespecified in the study design, given universal CD30 expression and known activity of Bv in this population [111]. These results led to FDA approval of Bv plus CHP for untreated CD30+ PTCL. Bv has also demonstrated activity as a single agent in R/R PTCL expressing CD30 [112,113]. Interestingly, Bv seems to be effective even in C30-negative T-cell lymphomas [114], and responses do not necessarily correlate with CD30 expression [115]. While the investigation is ongoing to better understand why Bv remains effective in CD30-negative disease, potential explanations include the bystander effect [116], lack of sensitivity when assessing for CD30 by IHC [117] and immunomodulation due to depletion of CD30+ regulatory T-cells [114,118]. These concepts, particularly the latter, were further supported by results from the Phase III randomized ECHELON-3 study presented at ASCO 2024, in which the addition of Bv to rituximab plus lenalidomide for R/R DLBCL led to significant benefits in PFS and OS, even among those with CD30-negative disease [119]. Brentuximab is now being studied in R/R T-cell lymphomas with low CD30 expression and in combination with either ICIs or mogamulizumab.

SGN-35T is a new CD30-directed MMAE ADC, which is similar to Bv but with a tripeptide cleavable linker consisting of D-leucine-alanine-glutamate (DLAE) found to reduce hematopoietic toxicity in preclinical investigation [120]. A Phase I study of SGN-35T in patients with CD30+ lymphomas is currently underway. Other ADCs and related targeted therapies under investigation for T-cell lymphomas include those targeting CD3, CD70, CD38, and TRBC1. Resimmune is a CD3-targeted immunotoxin consisting of catalytic and translocation domains of the diphtheria toxin fused to two anti-human CD3 Fv fragments [121]. Although three patients with PTCL were included in the Phase I clinical trial, there were no responses seen in these patients [122]. Resimmune remains under investigation with more promising results for CTCL. SGN-CD70A is an investigational ADC targeting CD70 with the cytotoxic DNA-crosslinking agent, pyrollobenzodiazepine (PBD) dimer, which has been studied in both renal cell carcinoma and non-Hodgkin lymphoma [123]. In a Phase I study of SGN-70A for diffuse large B-cell lymphoma (DLBCL) and mantle cell lymphoma (MCL), SGN-70A led to modest response rates limited by thrombocytopenia [124]. SGN-70A has also been studied in patient-derived xenograft CTCL models [125]. Its role in PTCL and potential candidacy for clinical trials remains to be determined. CD38-SADA is a targeted radiotherapy delivering fixed doses of ^177^Lu-DOTA, a chelated complex of a lutetium radioisotope and dotatate, which leads to targeted emission of ionizing beta radiation against CD38+ malignant cells, resulting in DNA damage and associated cancer cell death. Because a subset of lymphomas express CD38, including the majority of ENKTL cases, targeting CD38 with radioisotopes could provide both diagnostic and therapeutic benefits [126]. A Phase I clinical trial studying CD38-SADA in non-Hodgkin lymphomas is now underway. As with CARs, components of the TCR can also be targeted with ADCs. Nichakawade et al. developed a TRBC1-targeted ADC, which demonstrated potent anti-tumor properties using two different payloads (MMAE and tesirine) in mouse models with TRBC1+ T-cell cancer (Jurkat) cells [103]. This approach has not yet been evaluated in humans. Current studies using ADCs for PTCL are outlined in Table 4.

## 6. Conclusions

Antibodies provide unique specificity which can be leveraged for targeting the destruction of malignant cells with a variety of therapeutic modalities: mAbs, BsAs, CARs, and ADCs. Here we review current and investigational antibody-based therapies for the treatment of PTCL, as well as the advantages and disadvantages of each. Despite a limited number of approved agents, antibody-based therapies are essential tools for the treatment of PTCL and an increasing focus of investigation. Nonetheless, a few challenges unique to T-cell malignancies must be overcome for the development of safe and effective targeted therapies. While some of these therapies (e.g., CAR-T and TCEs) are inherently at risk for fratricide or bidirectional killing of malignant T-cells and effector T-cells, others are at risk for serious infectious complications associated with targeting pan T-cell antigens. Fortunately, recent preclinical investigation has revealed potential solutions through the identification of novel targets, combinations, therapeutic sequencing, and clever gene editing techniques. Following their dramatic impact on the treatment of B-cell malignancies, CAR therapies now represent the overwhelming majority of antibody-based therapies in clinical trials for PTCL and other T-cell malignancies. While it remains to be seen which of these novel antibody-based approaches will improve outcomes in PTCL, the future looks a little brighter for our patients.

## Figures and Tables

**Figure 1 cancers-16-03489-f001:**
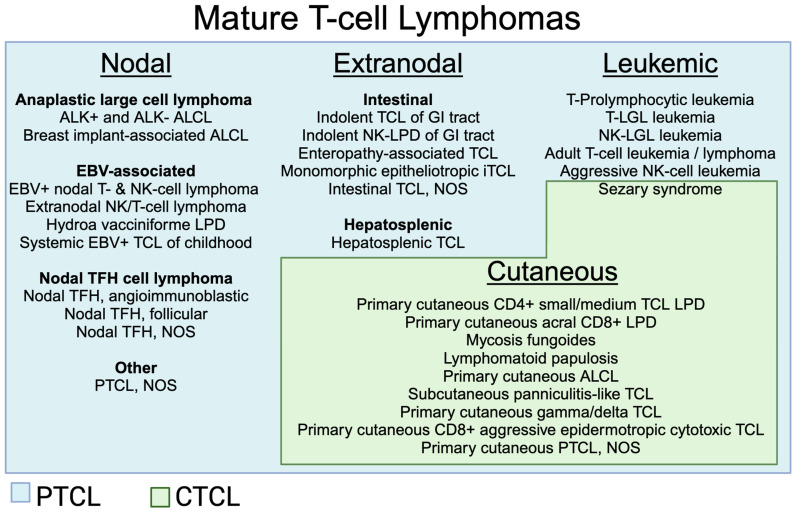
Subtypes of mature T-cell lymphomas. Abbreviations: ALCL (anaplastic large cell lymphoma), EBV (Epstein–Barr virus), iTCL (intestinal T-cell lymphoma), LGL (large granular lymphocytic), LPD (lymphoproliferative disorder), NOS (not otherwise specified), PTCL (peripheral T-cell lymphoma), TCL (T-cell lymphoma), TFH (T-follicular helper). Created using BioRender.

**Figure 2 cancers-16-03489-f002:**
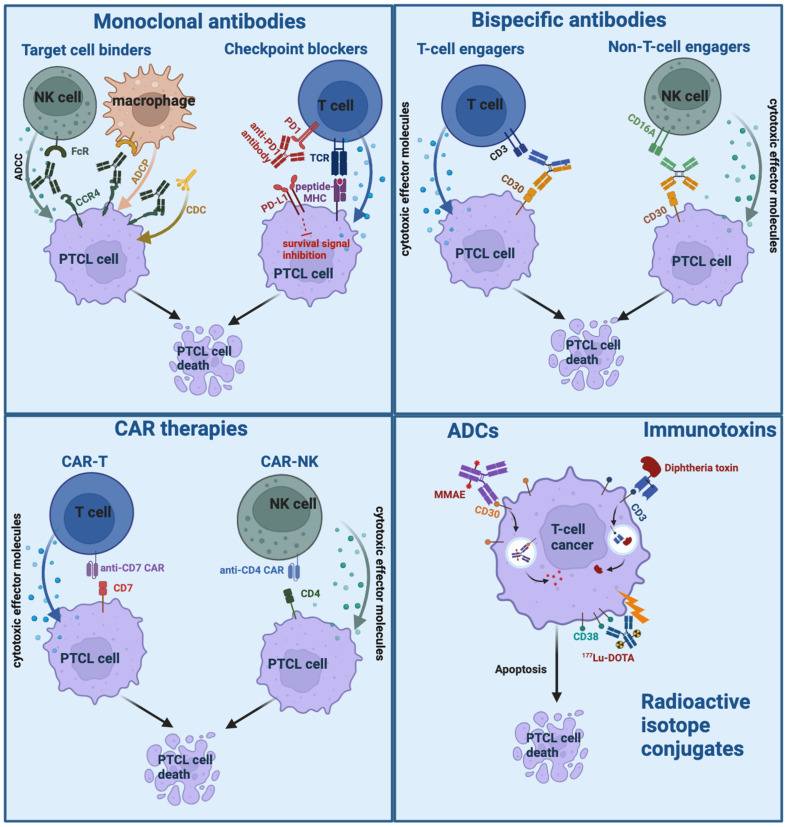
Mechanisms of antibody-based therapies with selected examples for peripheral T-cell lymphoma. Created using BioRender.

**Table 1 cancers-16-03489-t001:** Current studies using monoclonal antibodies in peripheral T-cell lymphoma.

Study ID	Title	Phase	Status	Sponsor
NCT04365036	A Multicenter, Phase III, Randomized Trial of Sequential Chemoradiotherapy With or Without Toripalimab (PD-1 Antibody) in Newly Diagnosed Early-Stage Extranodal Natural Killer/T-Cell Lymphoma, Nasal Type (ENKTL)	3	Recruiting	Sun Yat-sen University
NCT05700448	Study of Sugemalimab (or Placebo) Plus PGemOx Regimen in Participants with Extranodal NK/T-Cell Lymphoma	3	Not yet recruiting	CStone Pharmaceuticals
NCT05254899	Anti-PD-1 Antibody and P-GEMOX Chemotherapy Combined with Radiotherapy in High-risk Early-Stage ENKTL	2	Recruiting	Chinese Academy of Medical Sciences
NCT01703949	Brentuximab Vedotin with or without Nivolumab in Treating Patients with Relapsed or Refractory CD30+ Lymphoma	2	Recruiting	University of Washington
NCT05821192	Chemotherapy Plus PD-1 Monoclonal Antibody in the Treatment of Refractory or Relapsed Peripheral T-Cell Lymphoma	2	Recruiting	The First Hospital of Jilin University
NCT05182957	Clinical Study of Anti-PD-1 Plus Lenalidomide and Azacitidine in Relapsed/Refractory Peripheral T-Cell Lymphoma	2	Recruiting	The First Affiliated Hospital of Soochow University
NCT04127227	Sintilimab with P-GemOx Regimen for Newly Diagnosed Advanced Extranodal Natural Killer/T-Cell Lymphoma, Nasal Type	2	Recruiting	Sun Yat-sen University
NCT04414969	Anti-PD-1 Antibody Combined with Peg-Asparaginase and Chidamide for the Early Stage of NK/T-Cell Lymphoma	2	Recruiting	Hunan Cancer Hospital
NCT04984837	Study of Lacutamab in Peripheral T-Cell Lymphoma	2	Recruiting	The Lymphoma Academic Research Organisation
NCT04763616	Study of Isatuximab and Cemiplimab in Relapsed or Refractory Natural Killer/T-Cell Lymphoma Malignancy (ICING)	2	Recruiting	Samsung Medical Center
NCT05996185	Study of Mogamulizumab With DA-EPOCH in Patients With Aggressive T-Cell Lymphoma	2	Not yet recruiting	Yale University
NCT05475925	A Study of DR-01 in Subjects With Large Granular Lymphocytic Leukemia or Cytotoxic Lymphomas	1/2	Recruiting	Dren Bio
NCT06376721	Linperlisib Combined with Camrelizumab and Pegaspargase in Advanced or Relapsed/Refractory NK/T-Cell Lymphoma	1/2	Recruiting	Beijing Tongren Hospital
NCT03598998	Pembrolizumab and Pralatrexate in Treating Patients with Relapsed or Refractory Peripheral T-Cell Lymphomas	1/2	Not yet recruiting	City of Hope
NCT03011814	Durvalumab With or Without Lenalidomide in Treating Patients with Relapsed or Refractory Cutaneous or Peripheral T-Cell Lymphoma	1/2	Not yet recruiting	City of Hope
NCT04848064	Third-Party Natural Killer Cells and Mogamulizumab for the Treatment of Relapsed or Refractory Cutaneous T-Cell Lymphomas or Adult T-Cell Leukemia/Lymphoma	1	Recruiting	Ohio State University
NCT06385522	A Clinical Trial in Adults With Non-Hodgkin Lymphoma (NHL), With a Particular Emphasis on Cutaneous T-Cell Lymphoma (CTCL), Testing the Safety and Activity of a Novel Drug to Inhibit a Protein Called Tumor Necrosis Factor Receptor 2 That Drives Both Lymphoma Growth and Escape of the Immune System	1	Not yet recruiting	Boston Immune Technologies and Therapeutics
NCT02520791	Anti-ICOS Monoclonal Antibody MEDI-570 in Treating Patients with Relapsed or Refractory Peripheral T-Cell Lymphoma Follicular Variant or Angioimmunoblastic T-Cell Lymphoma	1	Not yet recruiting	National Cancer Institute

**Table 2 cancers-16-03489-t002:** Current studies using bispecific antibodies in peripheral T-cell lymphoma.

Study ID	Title	Phase	Status	Sponsor
NCT05883449	Phase II Study of AFM13 in Combination with AB-101 in Subjects With R/R HL and CD30+ PTCL (LuminICE-203)	2	Recruiting	Affimed Gmb
NCT05627856	A Study of GNC-038 Injection in Patients with Relapsed or Refractory NK/T-Cell Lymphoma, AITL, and Other NHL	1/2	Recruiting	Sichuan Baili Pharmaceutical Co., Ltd.
NCT05079282	Study of ONO-4685 in Patients with Relapsed or Refractory T-Cell Lymphoma	1	Recruiting	Ono Pharmaceutical Co., Ltd.
NCT05544968	CD30biAb-AATC for CD30+ Malignancies	1	Not yet recruiting	Medical College of Wisconsin

**Table 3 cancers-16-03489-t003:** Current studies using chimeric antigen receptor therapies in peripheral T-cell lymphoma.

Study ID	Title	Phase	Status	Sponsor
NCT05941156	Clinical Study of Anti-CD56-CAR-T in the Treatment of Relapsed/Refractory NK/T Cell Lymphoma/NK Cell Leukemia	2	Recruiting	The Affiliated Hospital of Xuzhou Medical University
NCT03590574	Phase I/II Study Evaluating AUTO4 in Patients With TRBC1 Positive T-Cell Lymphoma	1/2	Recruiting	Autolus Limited
NCT06492304	A Safety and Efficacy Study Evaluating CTX131 in Adult Subjects With Relapsed/Refractory Hematologic Malignancies	1/2	Recruiting	CRISPR Therapeutics AG
NCT06420089	CD5-deleted Chimeric Antigen Receptor Cells (Senza5 CART5) for T-Cell Non-Hodgkin Lymphoma (NHL)	1	Recruiting	Vittoria Biotherapeutics
NCT05377827	Dose-Escalation and Dose-Expansion Study to Evaluate the Safety and Tolerability of Anti-CD7 Allogeneic CAR T-Cells (WU-CART-007) in Patients With CD7+ Hematologic Malignancies	1	Recruiting	Washington University
NCT05290155	Anti-CD7 CAR T-Cell Therapy for Relapse and Refractory CD7 Positive T-Cell Malignancies	1	Recruiting	Shanghai General Hospital
NCT04288726	Allogeneic CD30.CAR-EBVSTs in Patients With Relapsed or Refractory CD30-Positive Lymphomas	1	Recruiting	Baylor College of Medicine
NCT04083495	CD30 CAR for Relapsed/Refractory CD30+ T-Cell Lymphoma	1	Recruiting	UNC Lineberger Comprehensive Cancer Center
NCT03829540	CD4CAR for CD4+ Leukemia and Lymphoma	1	Recruiting	Indiana University
NCT03690011	Cell Therapy for High-Risk T-Cell Malignancies Using CD7-Specific CAR Expressed On Autologous T-Cells	1	Recruiting	Baylor College of Medicine
NCT03081910	Autologous T-Cells Expressing a Second Generation CAR for Treatment of T-Cell Malignancies Expressing CD5 Antigen	1	Recruiting	Baylor College of Medicine
NCT02917083	CD30 CAR T-Cells, Relapsed CD30 Expressing Lymphoma (RELY-30)	1	Recruiting	Baylor College of Medicine
NCT05995028	Universal 4SCAR7U Targeting CD7-Positive Malignancies	1	Recruiting	Shenzhen Geno-Immune Medical Institute
NCT05620680	CD7 CAR T-Cells in T-Cell Lymphoma/Leukemia	1	Recruiting	Shenzhen University General Hospital
NCT06176690	Constitutive IL7R (C7R) Modified Banked Allogeneic CD30.CAR EBVSTS for CD30-Positive Lymphomas	1	Not yet recruiting	Baylor College of Medicine
NCT04712864	Study of CD4-Targeted Chimeric Antigen Receptor T-Cells (CD4- CAR-T) in Subjects With Relapsed or Refractory T-Cell Lymphoma	1	Not yet recruiting	Legend Biotech USA Inc
NCT06345027	Chimeric Antigen Receptor Treatment Targeting CD70 (Seventy) (Casey)	1	Not yet recruiting	Baylor College of Medicine
NCT04526834	Phase I Study of Autologous CD30.CAR-T in Relapsed or Refractory CD30 Positive Non-Hodgkin Lymphoma	1	Not yet recruiting	Tessa Therapeutics
NCT04502446	A Safety and Efficacy Study Evaluating CTX130 in Subjects With Relapsed or Refractory T- or B-Cell Malignancies (COBALT-LYM)	1	Not yet recruiting	CRISPR Therapeutics AG
NCT05979792	Clinical Study of CD7 CAR T-Cell Injection in the Treatment of Patients With Relapsed or Refractory CD7-Positive Peripheral T-Cell Lymphoma	1	Not yet recruiting	Ruijin Hospital
NCT05013372	CD147-CAR T-Cells for Relapsed/Refractory T-Cell Non-Hodgkin Lymphoma	1	Not yet recruiting	Peking University People’s Hospital

**Table 4 cancers-16-03489-t004:** Current studies using antibody–drug conjugates in peripheral T-cell lymphoma.

Study ID	Title	Phase	Status	Sponsor
NCT02588651	A Phase II Study of Single Agent Brentuximab Vedotin in Relapsed/Refractory CD30 Low (<10%) Mature T-cell Lymphoma (TCL)	2	Recruiting	Cleveland Clinic, Case Comprehensive Cancer Center
NCT05313243	Pembrolizumab and Brentuximab Vedotin in Subjects with Relapsed/Refractory T-Cell Lymphoma	2	Recruiting	Yale University
NCT05316246	Efficacy and Safety of BV with Tislelizumab for the Treatment of CD30+ Relapsed/Refractory NK/T-Cell Lymphoma	2	Not yet recruiting	Shanghai Zhongshan Hospital
NCT06120504	A Safety Study of SGN-35T in Adults with Advanced Cancers	1	Recruiting	Seagen Inc.
NCT05994157	Phase I, Open-label, Dose-escalation Trial with CD38-SADA:177 Lu-DOTA Drug Complex in Patients with Relapsed or Refractory Non-Hodgkin Lymphoma	1	Not yet recruiting	Y-mAbs Therapeutics

## Data Availability

Data sharing is not applicable.

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
