# Peer review of "Antibody-Based Therapies for Peripheral T-Cell Lymphoma"

_cancers, 2024, doi:10.3390/cancers16203489_

Round 1
Reviewer 1 Report
Comments and Suggestions for Authors
Dear authors,
Congratulations on your valuable work! The topic is one of great interest. This comprehensive review provides a thorough, yet critical presentation of all relevant literature data. The manuscript is well organized, the style is clear and concise. The figures and tables make the data more accessible.
However, I have a few suggestions.
1. Please consider including a brief presentation of the classification of PTCLs and a diagram illustrating the PTCL subtypes.
2. “In addition to disease rarity and heterogeneity, the delayed progress in PTCL is driven in part by fewer targets and limited effective treatment options.” (lines 31 and 32) Please rephrase for better clarity.
3. “ In addition to their role as immune checkpoint inhibitors, monoclonal antibodies exhibit antitumoral activity through direct binding of the target cell via the Fab region of the antibody and engagement of immune cells—namely, natural killer (NK) cells and macrophages—through the Fc region, ultimately leading to antibody-dependent cellular cytotoxicity (ADCC), antibody-dependent cellular phagocytosis (ADCP), and complement-dependent cytotoxicity (CDC).” (lines 57-63)
Please consider improving the discussion regarding the mechanisms of action of monoclonal antibodies, which may also induce direct apoptosis or block chemokine-mediated cellular migration and proliferation of T cells, chemokine-mediated angiogenesis.
4. Anti-CD30 antibodies should also be presented in the “Monoclonal antibodies” section.
Best regards!
Author Response
Thank you for pointing out comments 1-4. All changes have been made and integrated into the paper. Thank you for your time and thoughtful review
Reviewer 2 Report
Comments and Suggestions for Authors
In the manuscript „Antibody-Based Therapies for Peripheral T-Cell Lymphoma“ the authors provide an overview on antibody-based strategies for the treatment of peripheral T-cell lymphoma (PTCL) including monoclonal antibodies, bispecific antibodies, chimeric antigen receptor T-cell therapy and antibody-drug conjugates (ADCs) and discuss the limitations as well as potential future directions.
This review is well-written and well-structured, and provides up-to-date information on current and evolving strategies and lists current studies in tables. However, some minor points might be addressed to further improve the manuscript.
- Figure 1/monoclonal antibodies: The main mechanism of checkpoint blockade is to abrogate T-cell inhibition. In addition, the illustration also addresses the inhibition of survival signaling in malignant cells by disrupting the PD-PD-L1 axis. This additional mechanism of action should thus also be explained in the respective section of the main text.
- Figure 1: Cell-mediated cytotoxicity relies on the concerted action of different effector molecules with perforin and granzyme B being the most prominent ones. However, depending on the effector cell type other cytotoxins like granulysin or additional granzymes might contribute to cytotoxicity. Thus, only displaying „perforin“ in the figure might be a bit oversimplified and I suggest using a more generalized term like „cytotoxic effector molecules“.
- Line 413-415: The description of CD70 might be moved to section 3, where CD70 is mentioned first.
- Some minor typos including line 206 (comma missing between refs 41 and 42), 366 (CD3CAR), 367 (Jukrat), 398 (honing), 461 (in in)
Author Response
Thank you for your comments.
Comment 1: a Section on PDL1 activity has now been integrated into the paper, under section 2.5
Comment 2: Figure 1 has now been modified
Comment 3/4: Changes have been integrated
Reviewer 3 Report
Comments and Suggestions for Authors
The manuscript of Nazila Shafagati, Suman Paul, Sima Rozati and Cole H. Sterling is devoted to a description of modern approaches to antitumor immunotherapy for peripheral T-Cell lymphoma (PTCL) using various types of antibody-based protocols. This area of research relates to very significant problems in oncology, because “PTCL represents a rare and heterogeneous subset of non-Hodgkin lymphomas with inferior outcomes for most subsets”. In the literature review, the authors “explore four areas of current and evolving antibody-based strategies for the treatment of PTCL: monoclonal antibodies (mAbs), bispecific antibodies (BsAs), chimeric antigen receptor T-cell therapy (CAR-T), and antibody-drug conjugates (ADCs)”. Along with a detailed description of the features of these approaches, the authors of this literature review discuss their advantages and disadvantages. The main focus of the manuscript is on analyzing the effectiveness of using antibodies, T lymphocytes, NK cells and macrophages in PTCL immunotherapy. It is very important for this review that the authors consider various protocols for the use of these approaches in antitumor therapy, especially for PTCL, including options for using a number of relevant genetic and other modifications of the immune cells used. The significance of this review is beyond doubt, since the presented summary of literature data and their analysis give an objective integral picture of existing ideas on the indicated topic and highlight possible ways of practical use of the accumulated knowledge. This includes very informative and interesting sections of the manuscript, in particular, Monoclonal Antibodies, Bispecific Antibodies, CAR T-Cell Therapies, Antibody-Drug Conjugates. The figure and four tables given in the manuscript adequately illustrate its main content. The final part of the manuscript “Conclusions” summarizes the reasonable arguments for the conclusion.
In general the review is well presented; the data are of considerable novelty and interest. In fact, the manuscript is ready for publication, but several minor suggestions might improve the overall quality of the manuscript:
1. (line 83) The abbreviations PFS and OS (earlier line 97) should be expanded.
2. (line 104) The abbreviation SD should be expanded.
3. (line 133) The abbreviation CRR should be expanded.
4. (line 252) The abbreviation CB-derived should be expanded.
5. (line 295) The abbreviation CR should be expanded.
6. (line 320) The abbreviation TRAF should be expanded.
7. (line 360) The abbreviation PEBL should be expanded.
8. (line 366) “(N92)” should be substitute for “(NK-92)”.
9. (line 379) “N92” should be substitute for “NK-92”.
10. (line 392) The abbreviation cHL should be expanded.
11. (line 443) The abbreviation PR should be expanded.
12. (line 502) The abbreviation CHOP should be expanded.
Author Response
Thank you for all comments, 1-12. They have now been accepted and integrated.